# Identification and Validation of the lncRNA MYOSLID as a Regulating Factor of Necroptosis and Immune Cell Infiltration in Colorectal Cancer following Necroptosis-Related LncRNA Model Establishment

**DOI:** 10.3390/cancers14184364

**Published:** 2022-09-07

**Authors:** Zhiwei Wu, Fan Zhang, Yaohui Wang, Zhixing Lu, Changwei Lin

**Affiliations:** 1Department of Health Management, The Third XiangYa Hospital of Central South University, Changsha 410017, China; 2Department of Gastrointestinal Surgery, The Third XiangYa Hospital of Central South University, Changsha 410013, China; 3Department of Gastrointestinal, Hernia and Enterofistula Surgery, People’s Hospital of Guangxi Zhuang Autonomous Region, Nanning 530000, China

**Keywords:** necroptosis, lncRNA, colorectal cancer, tumour immunity, MYOSLID

## Abstract

**Simple Summary:**

Colorectal cancer is one of the most common cancers and the second leading cause of deaths due to cancer. In this study, we developed a neural model based on only four lncRNAs to predict the overall survival rate of colorectal cancer patients. Moreover, we validated the value of analysing the lncRNA MYOSLID, one of the hub lncRNAs in our model, which promotes colorectal cancer by regulating necroptosis. Our study offered some essential insights into predicting the prognosis of colorectal cancer patients and may help to assist diagnosis and treatment in the future.

**Abstract:**

Necroptosis is a newly defined form of programmed cell death that plays an important role in cancers. However, necroptosis-related lncRNAs (NRLs) involved in colorectal cancer (CRC) have not yet been thoroughly studied. Methods: In this study, a 4-NRL model was developed based on the least absolute shrinkage and selection operator (LASSO) algorithm. A series of informatic, in vitro and in vivo analyses were applied to validate the prognostic value of the model and the potential function of the hub lncRNA MYOSLID. Results: The model exhibited an excellent capacity for the prediction of overall survival and other clinicopathological features of CRC patients using Kaplan–Meier (K–M) survival curves and receiver operating characteristic (ROC) curves. Furthermore, a significant difference in the levels of immune cells, such as CD4 memory T cells and activated mast cells, between two risk groups was observed. The low-risk patients had a higher expression of immune checkpoints, such as PDCD1 (PD-1) and CD274 (PD-L1). The levels of MYOSLID, a hub lncRNA in our model, were higher in CRC tissues than in normal tissues. Knockdown of MYOSLID induced necroptosis and inhibited the proliferation of CRC cells in vitro and in vivo. Interestingly, knockdown of MYOSLID also increased the percentage of CD4^+^ and CD8^+^ T cells in subcutaneously transplanted tumours. Conclusion: Our model is a promising biomarker that can be used to predict clinical outcomes in CRC patients, and MYOSLID plays an important role in regulating necroptosis and immune cell infiltration in CRC.

## 1. Introduction

Colorectal cancer (CRC) has been reported to be the third most common diagnosed cancer and the second leading cause of cancer-related deaths in the world [1,2]. In 2020, over 1.9 million new cases were diagnosed. The increasing incidence of CRC is related to several factors, such as familial, sporadic and inherited factors [3,4]. Currently, many patients are diagnosed at an advanced stage with multiple symptoms, including haematochezia or colonic obstruction [2]. A recent study revealed that the overall 5-year relative survival rate for CRC patients is only 50% [5]. Additionally, tumorigenesis is extremely complex, with many immunologic factors involved in the process. So the identification of ideal novel biomarkers remains a high priority [6]. Although remarkable advances have been made in cancer immunotherapy in recent years and some immunotherapy targets such as PD-1, PD-L1 and CTLA4 have been clinically applied [7], it benefits only 3% to 7% of CRC patients who exhibit defective mismatch repair (dMMR) proteins/microsatellite instability (MSI-H) [8,9]. Thus, the majority of CRC patients cannot benefit from immunotherapy. Therefore, it is imperative to determine how immunotherapy can be augmented in most CRC patients, and it is essential to explore more novel biomarkers in the diagnosis and treatment of CRC.

Different kinds of cell death programs are essential as self-protection methods for the human body to remove abnormal cells such as cancer cells. However, many cancers, including CRC, often resist these cell death programs by abnormally expressing certain key factors, thus, maintaining the abnormal proliferation and metastasis of tumour cells. For example, apoptosis resistance has been considered a common phenomenon in many malignancies, which allows cancer cells to escape programmed cell death [10]. Recently, many types of programmed cell death, such as apoptosis [11], necrosis [12] and autophagy [13], have been indicated to be closely related to the sensitivity of patients to immunotherapy. They have become a substantial obstacle to modern cancer treatment. The urgency for deepening scientific understanding of the other regulatory roles of cell death processes, such as necroptosis, pyroptosis and ferroptosis, in tumour immunity is becoming increasingly apparent [14].

Necroptosis, which is a newly defined cell death process that differs from autophagy, apoptosis or necrosis, has been regarded as a promising alternative target to bypass the traditional cell death pathways [15]. Death in necroptotic cancer cells occurs by the stimulation of the inflammatory response or activation of the immunological response. Recent studies have revealed that regulators of necroptosis could be biomarkers for the prediction of prognosis in cancers [16]. Accumulating evidence indicates that CRC tumours inhibit the process of necroptosis by regulating many regulators, such as RIPK1, RIPK3, and MLKL [17,18]. More importantly, recent studies have noted that necroptosis is involved in cancer immunological pathways and affects the response of tumours to immunotherapy. For example, necroptosis could increase CD8^+^ leukocyte-mediated antitumor immunity by activating RIP1 and RIP3 [19]. ZBP1-MLKL necroptotic signalling enhances radiation-induced antitumor immunity [20]. Conversely, Seifert et al. found that RIP1 and RIP3 could promote pancreatic cancer progression through CXCL1- and Mincle-induced immune suppression [21]. Unfortunately, little is known about the mechanisms regulating necroptosis, and whether inducing necroptosis can improve the effectiveness of immunotherapy treatment in CRC remains to be further studied.

Long non-coding RNAs (lncRNAs) are transcripts longer than 200 nucleotides without translation capability, and they regulate gene expression at multiple levels [22]. Many lncRNAs have been identified to play important roles in CRC progression [23]. Accumulating evidence has shown that lncRNAs can protect tumour cells from necroptosis by downregulating the expression of some related proteins [24,25]. For example, LINC00176 was found to inhibit necroptosis in hepatocarcinoma by releasing certain miRNAs [26]. Therefore, the exploration of potential necroptosis-related lncRNAs (NRLs) in CRC would be helpful in shedding light on CRC immunotherapy. In fact, Liu et al. have already generated a prognostic necroptosis model for CRC [27]. However, some shortcomings of the current model remain. First, the model established by Liu et al. cannot be used as an independent prognostic factor. Second, no necroptosis-related lncRNAs have been thoroughly studied as potential biomarkers or therapeutic targets in CRC.

Herein, we aimed to identify NRLs in CRC and generate a clinically useful prognostic model. More importantly, we further selected the key NRL, lncRNA MYOSLID, for analysis and preliminarily explored its functions in vitro and in vivo. The expression of MYOSLID has been validated to be upregulated in hypoxic CRC cells [28], but the potential role of MYOSLID in regulating necroptosis has never been studied. Our study provides a reliable necroptosis regulatory factor and potential immunotherapeutic target for CRC.

## 2. Materials and Methods

### 2.1. Acquisition of Data

The RNA expression data and relevant clinical features of CRC patients (*n* = 514) were obtained from The Cancer Genome Atlas (TCGA), including data obtained from 473 tumour tissues and 41 adjacent normal tissues. The genecode.v22 file was applied to annotate Ensemble IDs into gene symbols. All data were then log2(x + 1) transformed after the FPKM value was converted into the TPM value. To avoid bias in this study, CRC patents were randomly divided into training and test groups in the same proportions, which ensured that there were no clinical differences between these two groups.

### 2.2. Differential Expression Analysis

The ‘limma’ package [29] was applied to identify the differentially expressed lncRNAs (DElncRNAs) between CRC tissues and adjacent normal tissues. The threshold of DElncRNAs was defined as |log 2(fold change)| > 1 and a false discovery rate (FDR) < 0.05.

### 2.3. Extraction of Necroptosis Genes

The eight necroptosis genes were extracted from the official pathway called ‘GOBP_NECROPTOTIC_SIGNALING_PATHWAY’ in MSigDB (http://www.gsea-msigdb.org/gsea/msigdb/, accessed on 16 January 2022) and included FADD, FAS, FASLG, MLKL, RIPK1, RIPK3, TLR3 and TNF. Additionally, we collected the identities of newly reported genes involved in the necroptosis process to form a large list. Finally, necroptosis genes were analysed for this study (Appendix A).

### 2.4. Identification of Necroptosis-Related lncRNAs

Spearman correlation analysis was conducted to define necroptosis-related lncRNAs (NRLs). The inclusion criteria were |cor| > 0.3 and *p* < 0.05.

### 2.5. Establishment of the Co-Expression Network

Cytoscape 3.7.2 software was used to generate a co-expression network. This network illustrates the correlation between NLR levels and the levels of their corresponding mRNAs. Additionally, a Sankey diagram was generated to demonstrate the specific roles of NLRs (risk/protective factors) and their connections with the correlated mRNAs.

### 2.6. Establishment of the Prognostic Model

The model was generated using the method described in our previous study [30]. Univariate Cox proportional hazard regression analysis was then implemented to screen these NRLs via the ‘survival’ package. The NRLs that significantly correlated with overall survival were used for LASSO [31] regression and 10-fold cross-validation. Finally, a model was formatted, and the patients’ risk scores were calculated by the formula below:$$\mathrm{Risk}\;\mathrm{score}=\sum_{1}^{n}\;\mathrm{coef}\left(\mathrm{NRL}\right)*\mathrm{expr}\left(\mathrm{NRL}\right)$$
where coef represents the coefficient of the NRLs generated from LASSO and expr represents the expression level of the corresponding NRLs. Next, the correlation between the risk score and the prognosis of patients was further estimated via Kaplan–Meier survival analysis and receiver operator characteristic curves [32]. Then, to assess the model reliability, we calculated the risk score in the validation dataset using the same formula, and the same assessment methods as above were conducted.

### 2.7. Functional Annotation and Pathway Enrichment Analysis

Functional annotation analysis in the GO and KEGG datasets was performed using the ‘clusterProfiler’ package [33]. The statistical significance standard was set at both *p* value < 0.05 and FDR < 0.05 simultaneously. Finally, bubble plots were drawn for visualization, which included three sections: biological processes (BP), cellular components (CC) and molecular functions (MF).

### 2.8. Gene Set Enrichment Analysis

GSEA was conducted to identify the differences in the molecular and biological processes between the high-risk and low-risk patients. We obtained both HALLMARK and KEGG gene sets from MSigDB (https://www.gsea-msigdb.org/gsea/msigdb, accessed on 16 January 2022). A *p* value < 0.05 and FDR < 0.25 were regarded as the inclusion criteria, which referenced the ‘clusterProfiler’ package manual. Single-sample GSEA (ssGSEA) was implemented with several representative gene sets using the ‘GSVA’ package [34].

### 2.9. Assessment of Immune Cell Infiltration and Cibersort Analysis

A deconvolution algorithm [35] was used to assess the immune cell infiltration levels in the CRC patients. Additionally, the difference in cell infiltration between the high-risk and low-risk patients was illustrated. Cibersort analysis (https://cibersort.stanford.edu/, accessed on 16 January 2022), which is an online tool that is used to estimate the relative proportion of multiple immune subsets based on a gene expression matrix, was performed. We planned to explore the differences in the proportions of immune cells between the high-risk and low-risk patients, as well as the relationship between the scores and immune cell abundance. The “ggplot2” package was used for visualization of the data.

### 2.10. Drug Sensitivity Prediction

To predict the drug response in CRC patients, the ‘Oncopredict’ package was used to calculate the IC_50_ values of multiple chemotherapy drugs. The IC_50_ value indicates the effectiveness of a certain substance in inhibiting a biological or biochemical processes. Additionally, Connectivity map (CMap) datasets were applied to investigate effective agonists or antagonists.

### 2.11. Tissue Sample Collection and Colon Cancer Cell Line Culture

All paired tissues were collected from the Gastrointestinal Surgery Department after passing the official audit of the Xiangya 3rd Hospital Medical Ethics Committee. All patients signed informed consent forms before surgery. A total of 40 pairs of tissues, including tumour and adjacent normal tissues, were obtained from CRC cancer patients who underwent tumour resection surgery and whose diagnoses were confirmed by pathological diagnosis between October 2020, and August 2021. All of the samples were maintained at −80 °C. Five cell lines, including human intestinal epithelial cells (FHCs), HCT116, HT29, SW480 and SW620, were purchased from ATCC, and all cells were cultured in culture media with 10% foetal bovine serum (Gibco BRL, Gaithersburg, MD, USA) at 37 °C, 95% humidity, and 5% CO_2_.

### 2.12. RNA Extraction and Quantitative Real-Time Polymerase Chain Reaction (qRT-PCR)

Total RNA from the tissues and cell lines was extracted by a Total RNA Extraction Reagent kit (10606ES60, Yeasen, Shanghai, China) according to standard protocols. Then, a cDNA synthesis kit (11139ES10, Yeasen, Shanghai, China) was used for reverse transcription. Gene expression was quantified by a Roche LightCycler 480 machine via SYBR Green Master Mix (11201ES03, Yeasen, Shanghai, China). The expression levels were calculated based on a semiquantitative method following the formula 2^−ΔΔct^. We chose GAPDH as the internal reference for normalization. All primers involved in this experiment were synthesized by Tsingke Biotech (Tsingke, Beijing, China). All the primers are listed in Appendix A.

### 2.13. In Vitro Experiments

Cell proliferation assays, such as the Cell Counting Kit-8 (CCK-8), colony formation, 5-ethynyl-20-deoxyuridine (EdU) and calcein-AM/PI assays, were conducted strictly following official protocols. All the details are provided in the Appendix A.

### 2.14. Mouse Tumour Models

Four- to six-week-old female BALB/c mice were purchased from the Department of Laboratory Animals of Central South University. CT26 tumour cells subjected to stable transfection in the MYOSLID knockdown and control groups and dissolved in 100 μL PBS were injected subcutaneously into syngeneic BALB/c mice (1 × 10^6^ cells in each). After twenty-one days, the mice were sacrificed, and the tumour weight and tumour volume were measured in each group. The tumour volumes were calculated following the formula: volume (mm^3^) = length (mm) × width (mm) × width (mm)/2.

### 2.15. In Vivo Experiments

Haematoxylin-eosin (H&E) staining, immunohistochemical assays and flow cytometry analysis were conducted strictly following official protocols. All the details are provided in the Appendix A.

### 2.16. Statistical Analysis

The proportion of tumour-infiltrating immune cells in the two groups was compared via the Wilcoxon test. The correlation between necroptosis gene levels and the levels of lncRNAs was calculated by Spearman correlation. The proportions of clinical features in the two groups were analysed by the chi-squared test. Univariate Cox and multivariate Cox regression analyses were applied to identify independent prognostic factors in CRC. Statistical significance was defined as *p* < 0.05, and all *p* values were two-tailed.

## 3. Results

### 3.1. Identification of Necroptosis-Related Differentially Expressed lncRNAs in CRC

Figure 1 depicts the overall research design of this study. A total of 514 samples with RNA-seq data were assessed using data obtained from the cancer genome atlas.

First, we aimed to identify validated necroptosis genes. The pathways extracted from MSigDB contained eight representative genes, namely, necroptosis-related genes (NRGs), including Fas Associated Via Death Domain (FADD), Fas Cell Surface Death Receptor (FAS), Fas Ligand (FASLG), Mixed Lineage Kinase Domain-Like Pseudokinase (MLKL), Receptor Interacting Serine/Threonine Kinase 1 (RIPK1), Receptor Interacting Serine/Threonine Kinase 3 (RIPK3), Toll-Like Receptor 3 (TLR3) and Tumour Necrosis Factor (TNF). A detailed description of these NRGs is provided in Appendix A. Subsequently, the distribution of these patients was illustrated using a PCA map, t-SNE plot and a bar chart (Appendix A). NRGs in the MSigDB database were used to identify necroptosis-related lncRNAs (NRLs). The relationship between the levels of these 8 NRGs and 14,086 lncRNAs was determined by Spearman correlation analysis. As a result, 1222 NRLs were identified when the inclusion criterion was set as a correlation coefficient (|R^2^|) > 0.3 and *p* value < 0.001. We also identified 1645 differentially expressed lncRNAs (DELs) (including 1239 upregulated DELs and 406 downregulated DELs) between CRC tissue and pericarcinous tissue using the CRC samples with the inclusion parameters log_2_|FC| > 1.5 and FDR < 0.05) (Figure 2A). Finally, we identified 421 necroptosis-related differentially expressed lncRNAs (NRDELs) (Figure 2B, Appendix A).

### 3.2. The Screening of Prognostic NRDELs in CRC Patients

To explore the NRDELs and their prognostic value, Cox univariate regression analysis was applied to identify the potential prognostically useful lncRNAs among all the NRDELs. Finally, 16 prognostically useful NRDELs (PNRDELs) were identified (Figure 2C, Appendix A). Fifteen PNRDELs were considered “risk” genes (HR > 1) and AC016027.1 was identified as a “protective” gene (HR < 1). The connection between the 16 PNRDELs and the NRGs was further assessed (Figure 2D, Appendix A).

We also visualized the prognostic function of these PNRDELs and aimed to explore an internal connection between the PNRDELs and NRGs by a Sankey diagram, which showed the relationship among the PNRDELs, NRGs and their functions in CRC (Figure 2E). The network of NRG and PNRDEL co-expression is shown in Appendix A. Among these necroptosis-related lncRNAs, the lncRNA FADD had the tightest linkage with necroptosis-related lncRNAs.

### 3.3. Generation and Validation of an NRL Prognostic Model

All of the CRC patients were randomly divided into two groups (the training group and the testing group), and the clinicopathological features of these two groups are presented in Table 1. All the *p* values describing the differences in these characteristics were higher than 0.05.

Based on the 16 PNRDEL data points mentioned above in the training group patients, a prognostic outcome evaluation model with 4 NRLs was then generated with the LASSO algorithm. The fit and lambda curves are shown in Figure 3A,B. Each CRC patient was endowed with a risk score by the formula Riskscore = MYOSLID × (1.20501) + ATP2A1AS1 × 0.12181 + AC016027.1 × (−0.28656) + AC010973.2 × 0.10369. We observed that lncRNA MYOSLID contributed the most to the score in this model. Cox univariate regression analysis showed that the risk score calculated by this formula was negatively correlated with CRC patient prognosis (*p* < 0.001; Figure 3C). Subsequently, multivariate Cox regression analysis also demonstrated that the newly defined 4-NRL model could function as an outstanding independent prognostic factor for assessing CRC patients (*p* < 0.001; Figure 3D). A nomogram also demonstrated the value of this risk-scoring system based on our model. The 1-year, 3-year and 5-year overall survival rates could be predicted ideally when our model was compared with the theoretical risk model (Figure 3E,F).

Moreover, we assessed the prognostic value of this 4-NRL model using several analyses. The CRC patients in both groups (training or testing group) were classified into low-risk and high-risk groups using the risk-scoring system. The risk scores (above) and the overall survival status (below) of CRC patients in both the high-risk and low-risk groups were distributed as expected (Figure 4A). As shown in Figure 4D, a K–M curve was plotted, showing that the overall survival rate of CRC patients in the low-risk group was significantly better than that in the high-risk group. The ROC curve resulting from the use of this model was also evaluated. The results showed that the areas under the curves (AUCs) were 0.699, 0.680 and 0.671 for analyses of 1-, 3- and 5-year survival, respectively (Figure 4G). A clinicopathological-related ROC curve was also generated and illustrated the outstanding predictive ability of this model for OS (Figure 4J). Next, distribution figures, heatmaps, K–M analysis and ROC analysis were applied to assess both the training group and the testing group. These analyses also demonstrated the high prognostic value of this model both in the training group (Figure 4B,E,H,K) and the testing group (Figure 4C,F,I,L). In summary, these results suggest that this model exhibits robust predictive accuracy for prognosis in the training, testing and overall groups.

### 3.4. The Relationship between the 4-NRL Model and Other Clinicopathological Features

We identified clinicopathological differences between the high- and low-risk groups. Significant differences were observed in stage (*p* < 0.01), pT stage (*p* < 0.001), pN stage (*p* < 0.001), pM stage (*p* < 0.05), venous invasion (*p* < 0.001) and lymphatic invasion (*p* < 0.05) between these two groups. Among the four NRLs in our model, MYOSLID, ATP2A1-AS1 and AC010973.2 were classified as high-risk lncRNAs, while AC016027.1 showed the opposite expression tendency (Figure 5A). The clinicopathological features were compared independently in Figure 5B–G. In summary, these results suggested that this 4-NRL model had excellent ability for predicting the outcome of CRC patients.

### 3.5. Discovery of Pathways and Functions of the New Model by GSEA and Gene Ontology Analysis

To investigate the possible signalling pathways involved in this 4-NRL model, gene-set enrichment analysis (GSEA) was applied. We observed that many cancer-related and immune-related pathways or hallmarks were enriched in high-risk groups, such as epithelial-mesenchymal transition, angiogenesis, allograft rejection, antigen prepossessing and presenting and intestinal immune network for IgA production. Importantly, we found that the TNFα pathway and inflammatory response, which were identified as closely related to necroptosis [36,37], were also activated in the high-risk group. Moreover, many metabolic pathways were suppressed in the high-risk group (Figure 6A, Appendix A).

To further explore the molecular processes that the molecules assessed in this risk model were involved in, Gene Ontology enrichment analysis was applied using the differentially expressed genes (DEGs) between the two risk groups. The details of the results are presented in Figure 6B. Altogether, cancer progression and cancer immunity pathways were significantly enriched for in our 4-NRL model.

**Figure 6 cancers-14-04364-f006:**
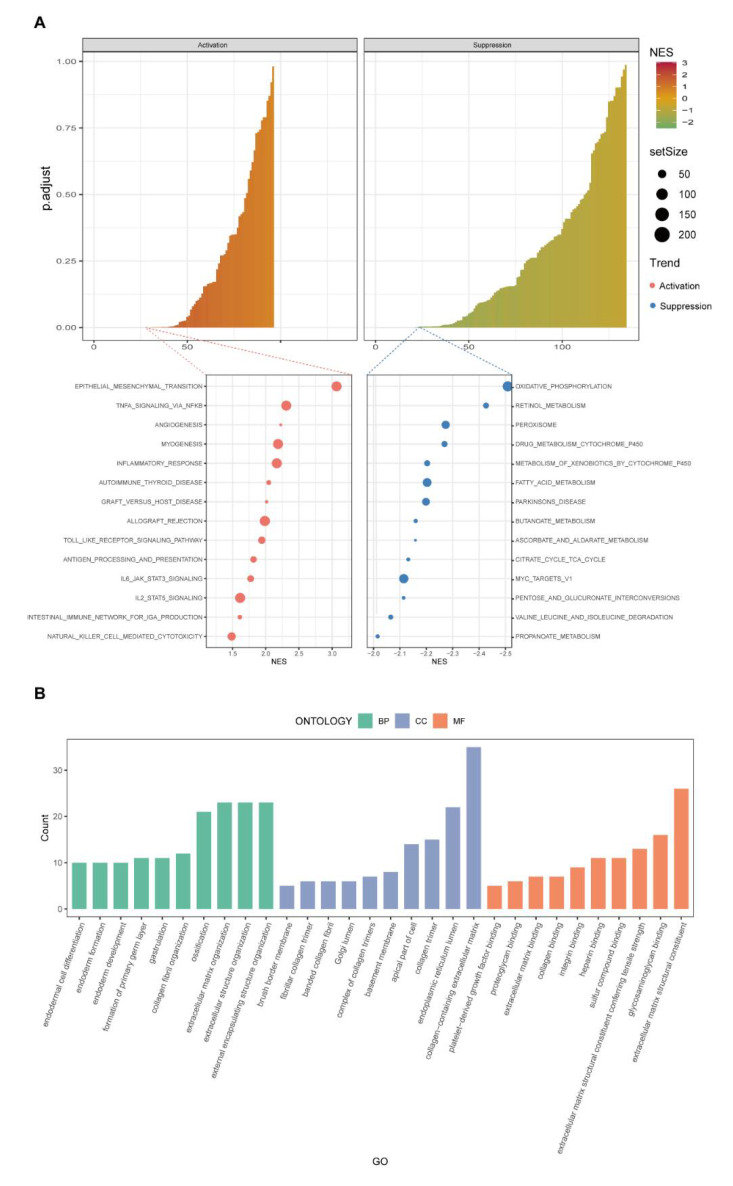
Pathway enrichment and gene ontology analyses of CRC patients based on the necroptosis−related lncRNA prognostic model. (**A**) GSEA revealed the activation of some immune−associated pathways and tumour progression pathways in high−risk CRC patients. (**B**) GO analysis revealed that biological processes (BP, green colour), cellular components (CC, blue colour) and molecular functions (MF, orange colour) were activated in the high−risk group of CRC patients.

### 3.6. Investigation of the Immunity Landscape Using the Established Model

We found that many immune-associated pathways or hallmarks were enriched in GSEA. Previous studies have also revealed that necroptosis is involved in tumour immunity or immunotherapy. Therefore, it is necessary to investigate the TME in CRC using our model. We observed that the levels of many immune cells were associated with the risk scores by a multiplatform immune cell correlation analysis (Figure 7A). We also found that a higher risk score was negatively associated with the levels of several immune cells, such as B plasma cells, CD4^+^ memory resting T cells and CD8^+^ T cells (Figure 7B–D, Appendix A). To further assess the reliability of our results, we identified differences in typical immune cells between the high- and low-risk groups and found that the levels of plasma cells, CD4 memory resting T cells, dendritic resting cells, mast resting cells, M0 macrophages and activated mast cells or neutrophils were significantly different between the two risk groups (Figure 7E). The expression levels of immune checkpoint genes were also compared between the high- and low-risk groups. Significant differences in the levels of 26 checkpoint genes between the high- and low-risk groups were observed. Twenty-two of them showed a higher expression level in the high-risk group. Among these, many validated immunotherapy targets, such as PDCD1 (PD-1), CD274 (PD-L1), and CTLA4, were involved. However, the expression of LGALS9 (Galectin-9), HHLA2, TMIGD2 (CD28H), and TNFRSF9 (CD137) was downregulated in the high-risk group (Figure 7F). In summary, these results indicated that the risk level of CRC patients classified by our model was associated with the levels of many infiltrating immune cells.

### 3.7. Exploration of Clinical Immune Treatment and Drug Sensitivity Tests in Two Risk Subgroups

To evaluate the immunotherapeutic outcome by risk scores, different IPS values (IPS-PD-1/PD-L1/PD-L2 positive and IPS-CTLA-4 positive) were applied as markers of the immunotherapy responses of CRC patients. Interestingly, the IPS–CTLA4 blocker score was significantly higher in the low-risk group (Figure 8A). However, we observed no difference in the outcome of individuals who used combination therapy (using combinative blocks: PD1/PDL1/PDL2 and IPS–CTLA4) and a single IPS–PD1/PDL1/PDL2 blocker (Figure 8B,C). Therefore, it is reasonable to conclude that CRC patients with lower risk scores might exhibit an effective response when treated with CTLA4 immunotherapy.

Moreover, higher IC_50_ values for some drugs for chemotherapy or targeted therapy, such as BIBW2992, were observed in the high-risk group. However, lower IC_50_ values for many candidate drugs, such as bleomycin, cyclopamine and DMOG, were also observed in the high-risk group, which indicates that these drugs might be better choices for this group. The 12 representative drugs with the lowest *p* values are shown in Figure 8D–O.

Taken together, these data demonstrate that many immune drugs might be functional in immune therapy in CRC patients.

### 3.8. The lncRNA MYOSLID Is a Regulatory Factor of Immune Infiltration in CRC

The LncRNA MYOSLID may function as a hub lncRNA in the novel 4-NRL model. To explore the functions of the oncogene MYOSLID, we analysed the expression level of MYOSLID in paired CRC samples using datasets from TCGA (N = 41). The results indicated that the MYOSLID expression was higher in most tumour samples than in non-tumor samples (Figure 9A). In addition, we compared the expression of MYOSLID among patients in different CRC stages and observed that its expression was upregulated in advanced CRC patients (Figure 9B). Moreover, time-dependent ROC analysis and Kaplan–Meier survival analysis of MYOSLID levels were applied to determine its prognostic value. The AUCs were more than 0.57 when the 1-, 3- and 5-year survival rates were assessed (Figure 9C). The overall survival rate was significantly lower in the high-risk group (*p* < 0.05, Figure 9D). We also observed a similar trend in the disease-free survival rate (DFS) (Figure 9E). Univariate/multivariate Cox regression analysis identified MYOSLID as a potential independent prognostic lncRNA in CRC (Appendix A).

Subsequently, the role of MYOSLID in tumour immunity was further investigated. We found that the expression level of MYOSLID was related to the levels of many well-known immune checkpoint genes, including CD80, LAIR1, CD28, PDCD1LG2, TNFSF4, CD160, TNFSF14, TNFSF9, HAVCR2, CD274 and CD86 (Figure 9F). We also confirmed an association between MYOSLID levels and immune cell infiltration levels using datasets from the CIBERSORT database. The results indicated that its expression level was correlated with the infiltration levels of 22 immune cells (Figure 9G). These results demonstrated that MYOSLID expression was upregulated in CRC and might have utility as a prognostic marker in CRC. In addition, it is a lncRNA that is potentially involved in immune infiltration in CRC.

**Figure 9 cancers-14-04364-f009:**
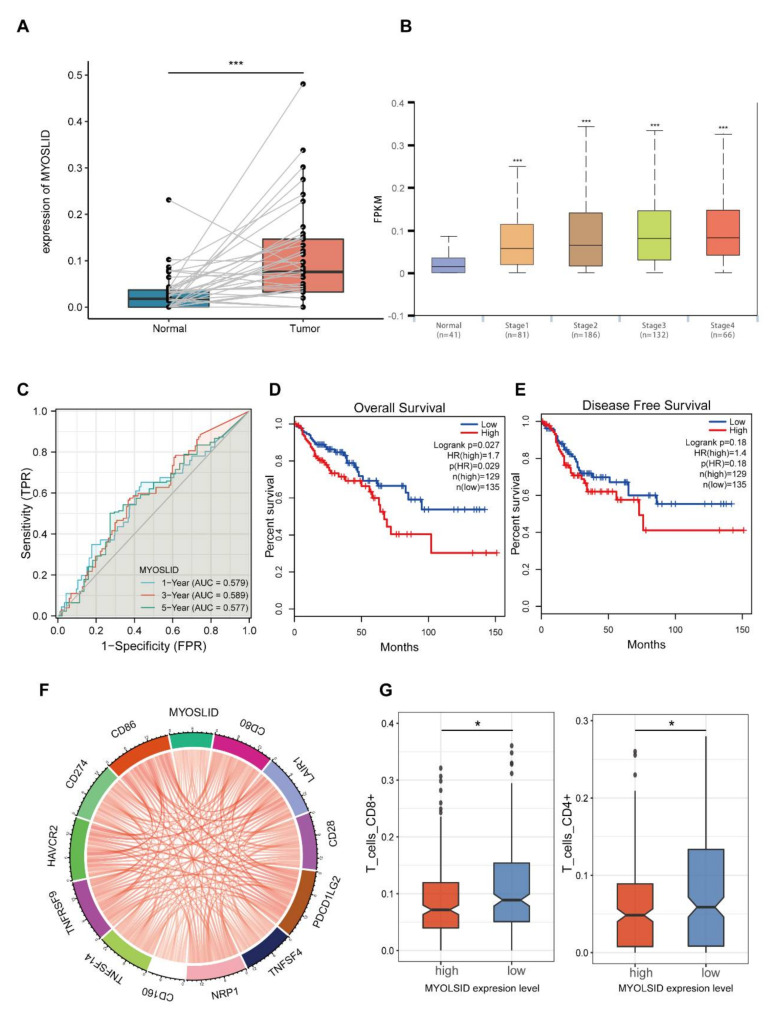
The expression and prognostic value of LncRNA MYOSLID and its association with immune infiltration in CRC patients. (**A**) The expression level of MYOSLID in CRC tissues and in pericarcinous tissues was investigated using datasets from the TCGA database. (**B**) MYOSLID expression levels in patients at different CRC stages. (**C**) The time−dependent ROC curve illustrates the potential use of MYOSLID in predicting the 1-, 3-, and 5-year overall survival rates of CRC patients. (**D**,**E**) The K–M analysis shows the correlation between MYOSLID expression level and the OS and DFS of CRC patients from the TCGA database. (**F**,**G**) The difference between the MYOSLID expression level and the levels of infiltrating CD4^+^ T/CD8^+^ T cells in CRC patients. *: *p* < 0.05, ***: *p* < 0.001, ns: no significance.

### 3.9. lncRNA MYOSLID Promotes the Growth of CRC by Inhibiting Necroptosis In Vitro

To investigate the function of the lncRNA MYOSLID in CRC, we first detected the expression level of MYOSLID in the FHC (human colon epithelial cell line) and four human CRC cell lines (LOVO, HCT116, RKO, SW480) (Figure 10A). Using samples from clinical CRC patients, we observed that MYOSLID levels were higher in most cancer samples than in their paired pericarcinous samples (Figure 10B). Then, we selected RKO (which had the highest expression level of MYOSLID) to establish a stable MYOSLID knockdown cell line, and we found that the knockdown level in shMYOSLID#1 was satisfactory (Figure 10C). We decided to use shMYOSLID#1 for follow-up research. CCK-8 analysis showed that the cell proliferation rate significantly decreased after the depletion of MYOSLID (Figure 10D). EdU assays also showed impaired proliferation in MYOSLID-knockdown cells (Figure 10E). Moreover, the colony-formation capacity of RKO cells was inhibited after the knockdown of MYOSLID (Figure 10F). These results proved that knockdown of MYOSLID impaired the growth of CRC cells.

Subsequently, we identified MYOSLID as a hub lncRNA in a necroptosis-related model. We explored its role in the necroptosis process. Several inhibitors of distinct signalling pathways (apoptosis, autophagy, necroptosis and ferroptosis) related to cell death were implemented to determine which types of cell death occurred after MYOSLID knockdown. As expected, we found that Q-VD-OPH (an inhibitor of the apoptosis pathway) and necrostatin-1 (a necroptosis inhibitor) partially rescued the growth ability of shMYOSLID RKO cells (Figure 10G). The calcein-AM/PI assay also demonstrated that necrostatin-1 partly rescued the proliferation rate of MYOSLID-depleted cells (Figure 10H). Collectively, these findings suggest that MYOSLID might promote the proliferation of CRC by inhibiting the necroptosis process.

**Figure 10 cancers-14-04364-f010:**
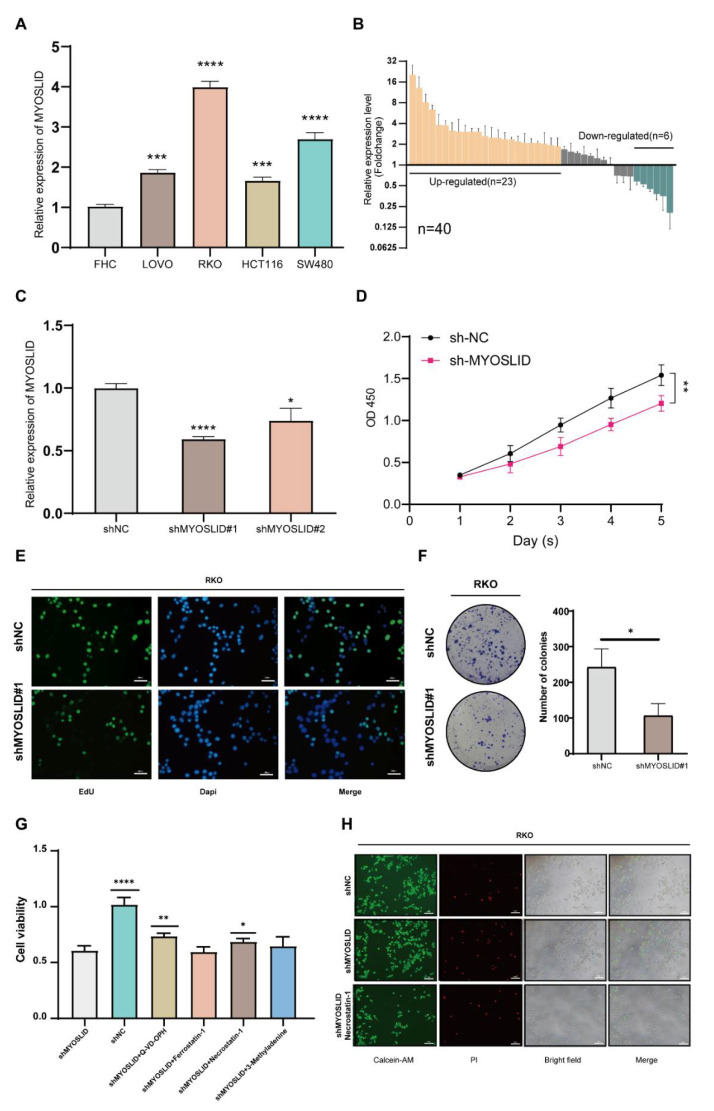
The LncRNA MYOSLID promotes the proliferation of CRC by regulating necroptosis. (**A**) A bar chart showing the expression level of MYOSLID in human normal colon epithelial cells (FHC) and four human CRC cell lines (LOVO, RKO, HCT116, SW480). (**B**) The expression level of MYOSLID in paired colorectal cancer samples. (**C**) The expression level of MYOSLID after knockdown by short hairpin RNA. (**D**) Cell proliferation was measured using CCK-8 assays in RKO cells. (**E**) Evaluation of cell proliferation in MYOSLID knockdown RKO cells by EdU assays. (**F**) Evaluation of the colony formation capacity in MYOSLID knockdown RKO cells. (**G**) The cell viability of the RKO wild-type group and the shMYOSLID stably transfected group after treatment with the blank control, Q-VD-OPH, necrostatin-1, ferrostatin-1 and 3-methyladenine. (**H**) The fluorescence intensity excited by Calcein-AM/PI indicated the viability of RKO cells after treatment with necrostatin−1. *: *p* < 0.05, **: *p* < 0.01, ***: *p* < 0.001, ****: *p* < 0.0001. ns: No significance.

### 3.10. The LncRNA MYOSLID Promotes CRC Growth and Inhibits the Infiltration of CD4^+^ and CD8^+^ T Cells In Vivo

A BALB/c mouse homograft was used to confirm the potential function of MYOSLID in vivo. The mouse CRC cell line CT26 was stably transfected with a shRNA targeting MYOSLID and subcutaneously injected into mice. We found that all mice developed tumours. Moreover, the tumour volume and tumour weight were significantly lower in the shMYOSLID group than in the control vector group (Figure 11A–E). The IHC staining results revealed that shMYOSLID CT26 tumour samples exhibited a higher expression of RIPK3 (Figure 11F). This finding indicates that necroptosis might be activated after MYOSLID knockdown. Moreover, flow cytometry showed that the levels of CD4^+^ T and CD8^+^ T cells were statistically higher in the MYOSLID knockdown group than in the control group (Figure 11G). In summary, MYOSLID may function in regulating the process of necroptosis and may also be involved in cancer immunity.

## 4. Discussion

Recently, necrosis was shown to influence the invasion and migration, as well as the immune reaction, of many types of cancers [2,38]. Some previous studies have explored the roles of lncRNAs in necroptosis [39,40]. Therefore, the attempt to identify more NRLs is no doubt important in cancer diagnosis and treatments. However, a previous study only generated a simple model and failed to demonstrate its reliable independent prognostic value in multivariate Cox regression analysis [25]. Studies of NRLs as well as an outstanding necroptosis-related model in CRC remain poorly developed.

In this study, we comprehensively analysed the expression characteristics of eight NRGs confirmed to be involved in the necroptosis pathway (RIPK3, MLKL, FAS, FASLG, TLR3, TNF, RIPK1, and FADD). Then, we identified some DENRLs. The prognosis of each CRC patient and the expression features of these NRLs were explored. Finally, we identified 16 candidate NRLs for further study.

After a lncRNA–mRNA co-expression network was implemented, we found that FADD was connected with 7 PNRLs in the network. FADD has been shown to play an important role in cancer and inflammation [41]. FADD has also been shown to regulate necroptosis as a necrosome [42]. However, the potential function of FADD in necroptosis is controversial. A study revealed that FADD-KO cells exhibited resistance to TNF-induced necroptosis [43]. However, another study claimed that FADD knockdown could induce necroptosis [44]. Therefore, more studies are needed to determine the specific mechanism of FADD in necroptosis.

Subsequently, a 4-NRL model was generated to assess its prognostic value in CRC patients. Patients were regrouped into low- and high-risk groups using the model, and many prognostic-related analyses were performed. Specifically, this model is more valuable in clinical use than many other published models because it incorporates the analyses of only 4 lncRNAs and exhibits a strong ability to predict the outcome of CRC patients; its predictive ability is comparable to that achieved using the traditional TNM grouping or AJCC stage. Many adverse clinical events, such as venous invasion or lymphatic node invasion, are easily predicted by this model in addition to predictions of the survival of CRC patients.

To assess how this model can be used to predict the progression of CRC, GSEA was applied. The results showed that epithelial mesenchymal transition, TNF signalling via NFKB and angiogenesis were activated in the high-risk group. In addition, many immune-associated pathways were activated in the high-risk group. Accumulating studies have found that necroptosis is closely correlated with cancer immunological pathways and cancer immunotherapy [45]. The relationship between necroptosis and immunosuppressive microenvironments has been repeatedly studied [46,47]. We observed that many immune-related hallmarks were enriched, such as antigen prepossessing and presenting and intestinal immune network for IgA production. Therefore, it is reasonable to assume that the molecules assessed in this newly established model are closely related to cancer immunity. GO analysis comprises three parts: BP, MF and CC. The results revealed the potential molecular function of these molecules as well. In conclusion, we found that necroptosis was relatively inhibited in the high-risk group through some immune pathways.

The two groups divided by our model had different immune microenvironments. We observed that the levels of many immune cells, such as plasma cells, CD4 memory resting T cells, dendritic resting cells, and mast resting cells, were significantly lower in the high-risk group. The levels of M0 macrophages and activated mast cells or neutrophils were significantly higher in the high-risk group. CD4 T-cell responses are essential in the cancer immune cycle and substantially influence the clinical outcome [48]. High densities of tumour-associated plasma cells have also been proven to improve the prognosis of breast cancer [49]. Therefore, we can infer that the function of CD4 and plasma cells in CRC patients was inhibited in the high-risk group. Moreover, a significant upregulation in activated mast cells and M0 macrophages was observed. Mast cells are also involved in the inflammatory response and tissue homeostasis in cancer as innate immune cells [50]. M0 macrophages can differentiate into M2 macrophages, and thus, function in macrophage-mediated immune modulation and macrophage-mediated drug delivery [51]. Multiplatform immune cell correlation analysis also revealed that cancer-associated fibroblasts (CAFs) were highly related to the risk score (based on our model) of CRC patients. Therefore, it is reasonable to infer that mast cells, tumour-associated macrophages (TAMs) or CAFs may be potential targets for necroptosis-related immunotherapy.

The activation of immune checkpoints was significantly different between risk groups. The expression of some immune checkpoint genes (PD-1, PD-L1, CTL4A, etc.) were lower in the low-risk group than in the high-risk group. Patients with higher risk scores may potentially benefit from immune checkpoint blockade [52]. A previous study used an extra consensus clustering analysis for tumour immunity. However, our 4-NRL model could be used to directly and independently predict the immune landscape of CRC patients based on risk scores. Thus, our model has further proven its superiority for use in CRC.

Additionally, it was obvious that the lncRNA MYOSLID played a decisive role in our 4-NRL model when the risk scores of CRC patients were calculated. MYOSLID was first found to promote vascular smooth muscle differentiation [53]. Recent studies revealed that MYOSLID could promote the development of gastric cancer [54], osteosarcoma [55] and head and neck squamous cell carcinoma [56]. However, how MYOSLID functions in CRC and what role MYOSLID plays in necroptosis have not yet been reported. Consistent with a previous report, our study revealed that the expression of MYOSLID was higher in CRC cells than in normal epithelial cells, and MYOSLID was also identified as a potential prognostic lncRNA in CRC. Additional in vitro assays illustrated that the knockdown of MYOSLID by shRNA inhibited the proliferation rate of CRC cells. MYOSLID levels showed a strong correlation with the necroptosis pathway. We discovered that necroptosis inhibitors could be used to rescue the growth inhibition of CRC cells by shMYOSLID. In vivo assays further demonstrated the potential of MYOSLID in regulating necroptosis. Therefore, these experiments suggest that MYOSLID may promote the growth of CRC cells by inhibiting necroptosis. Interestingly, we also observed that MYOSLID plays some roles in regulating CRC immunity since its knockdown may lead to a decrease in the levels of CD4^+^ T cells.

Our research has limitations, although many methods were utilized to establish our model, and this study provided novel insight into the role of necroptosis in CRC. Although some functional analysis was performed to investigate MYOSLID in our study, no other mechanistic research was conducted. Further research on the underlying mechanism is needed.

## 5. Conclusions

An unprecedented predictive prognostic model was established in our study using only 4 NRLs, and the relationship between our risk model and the immune landscape was explored. We also identified the hub lncRNA MYOSLID in our model and preliminarily ascertained the potential mechanism underlying how it regulates the development of colorectal cancer. These findings may offer some useful insights into predicting the prognosis of colorectal cancer and may help to assist in the diagnosis or treatment of colorectal cancer in the future.

## Figures and Tables

**Figure 1 cancers-14-04364-f001:**
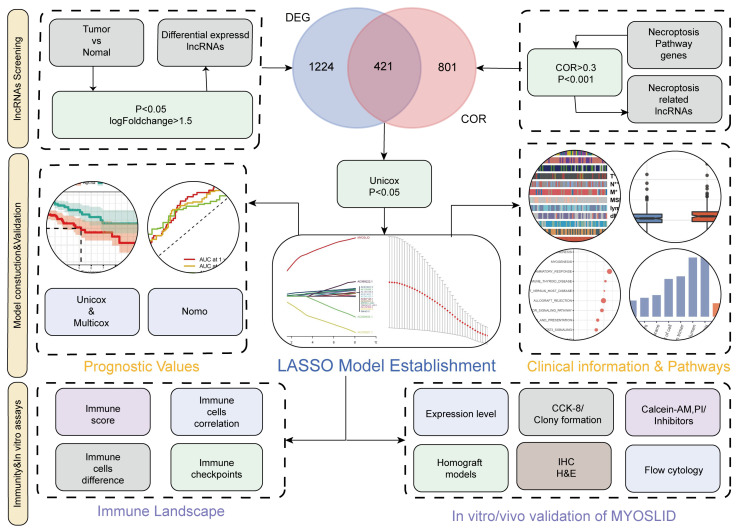
Study flowchart.

**Figure 2 cancers-14-04364-f002:**
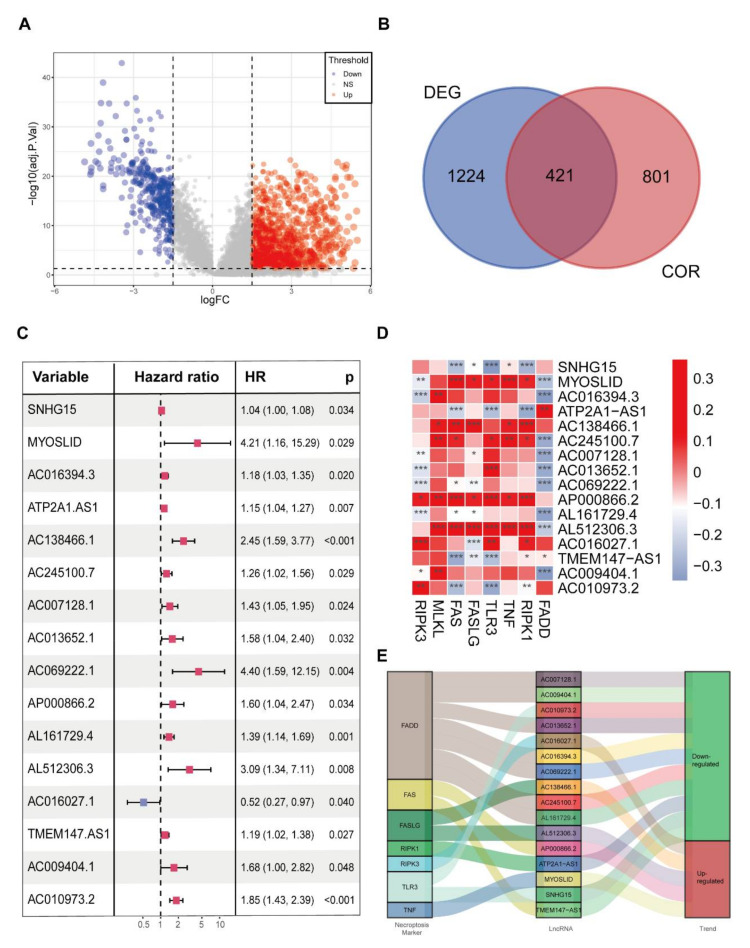
Prognostic evaluation of candidate necroptosis−related lncRNAs. (**A**) Volcano plots presenting the expression data of the 1645 differentially expressed lncRNAs in CRC patients from the TCGA−COAD cohort. (**B**) Venn diagram depicting the common collections of differentially expressed lncRNAs (DEGs) and necroptosis−related lncRNAs (CORs). (**C**) Cox univariate regression analysis of 16 candidate necroptosis−related lncRNAs. Red squire: Hazard ratio > 1, Blue squire: Hazard Ratio < 1(**D**) Heatmap showing the correlation between the levels of the 16 prognostic necroptosis−related lncRNAs and 8 validated hub genes in the necroptosis pathway in CRC patients. The colour of the blocks shows the degree of correlation. * *p* value < 0.05, ** *p* value < 0.01, and *** *p* value < 0.001. (**E**) The Sankey diagram presents the detailed connection between necroptosis−related lncRNAs and necroptosis−related genes.

**Figure 3 cancers-14-04364-f003:**
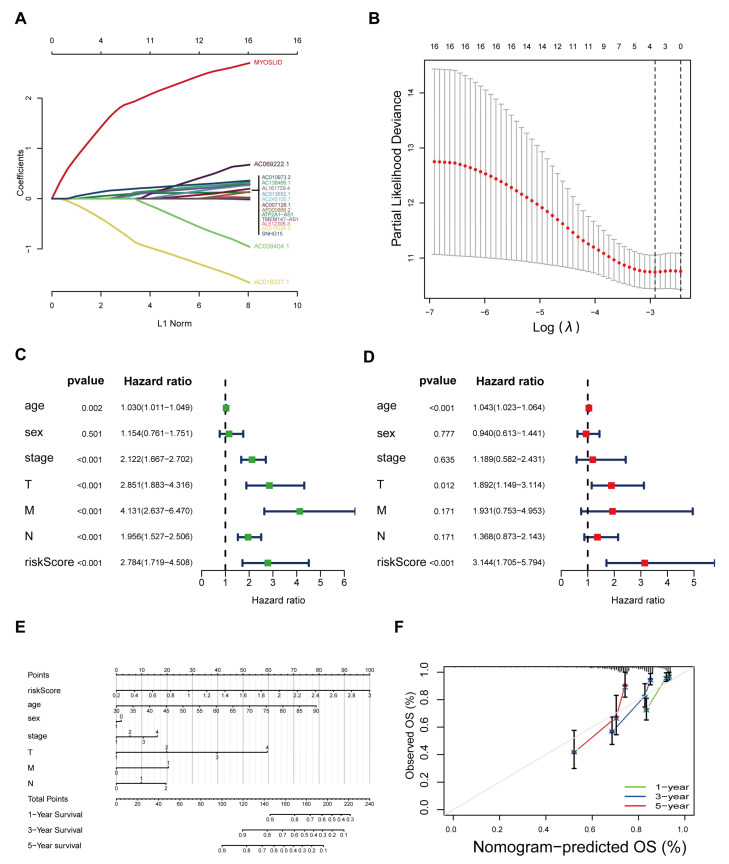
Establishment and preliminary prognostic validation of a 4-necroptosis-related lncRNA model. (**A**,**B**) Cvfit and lambda curves showing the correct criteria in the process of generating the model. (**C**,**D**) Univariate/multivariate Cox regression analysis comparing the prognostic value of the model with those achieved using other clinicopathological characteristics. (**E**) OS-related nomogram to predict the 1-, 3-, and 5-year OS rates in CRC patients. (**F**) Calibration curve to evaluate the predictive accuracy of the model. The grey diagonal line in the centre represents the ideal predictive situation.

**Figure 4 cancers-14-04364-f004:**
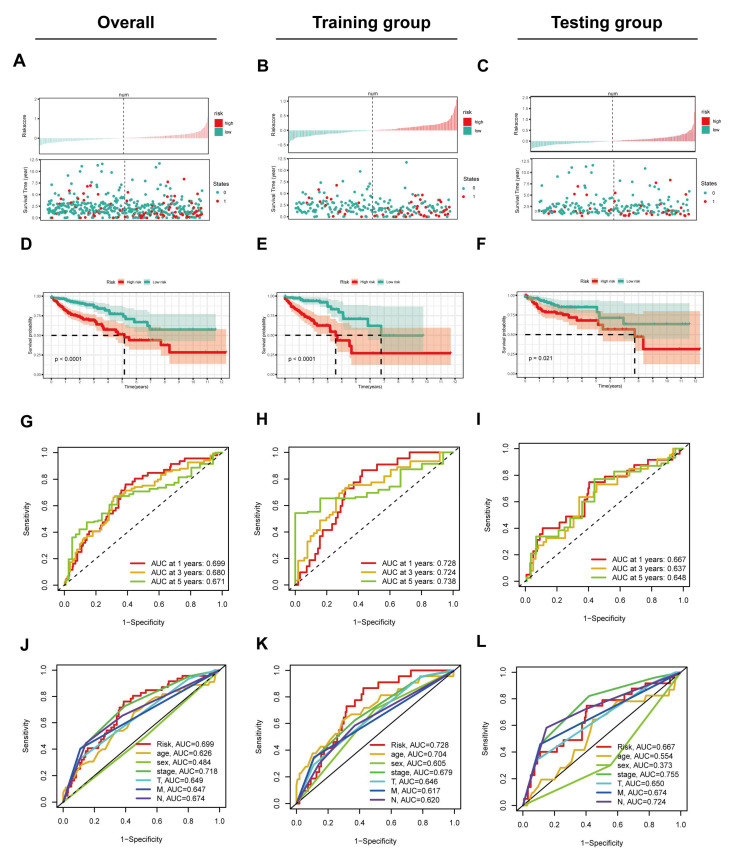
Validation of the necroptosis-related lncRNA model in the overall groups, the training group and the testing. (**A**–**C**) The distribution of the risk scores and the distributions of OS status in the overall, training and testing groups. (**D**–**F**) The K–M curves for overall survival in the overall, training and testing groups. (**G**–**I**) The ROC curve illustrates the prognostic value of the necroptosis-related lncRNA model in predicting the 1-, 3-, and 5-year OS rates in the overall, training and testing groups. (**J**–**L**) The ROC curve shows the prognostic value of the model and that of other clinicopathological factors in the overall, training and testing groups.

**Figure 5 cancers-14-04364-f005:**
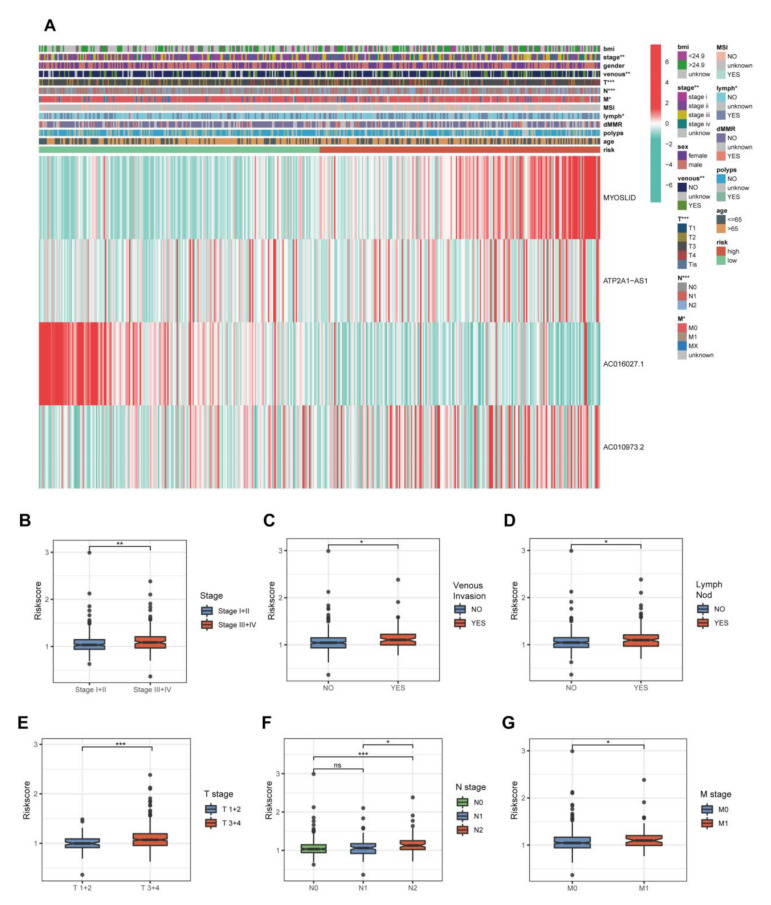
Relationship between the 4-NRL model and some representative clinicopathological features in CRC patients. (**A**) Heatmap showing the distribution of 12 representative clinicopathological features and the expression levels of 4 NRLs in CRC patients. (**B**–**G**) Histograms show the differences in the risk scores of CRC patients stratified by stage, T stage, N stage, M stage, lymph invasion and venous invasion. *: *p* < 0.05, **: *p* < 0.01, and ***: *p* < 0.001. ns: No significance.

**Figure 7 cancers-14-04364-f007:**
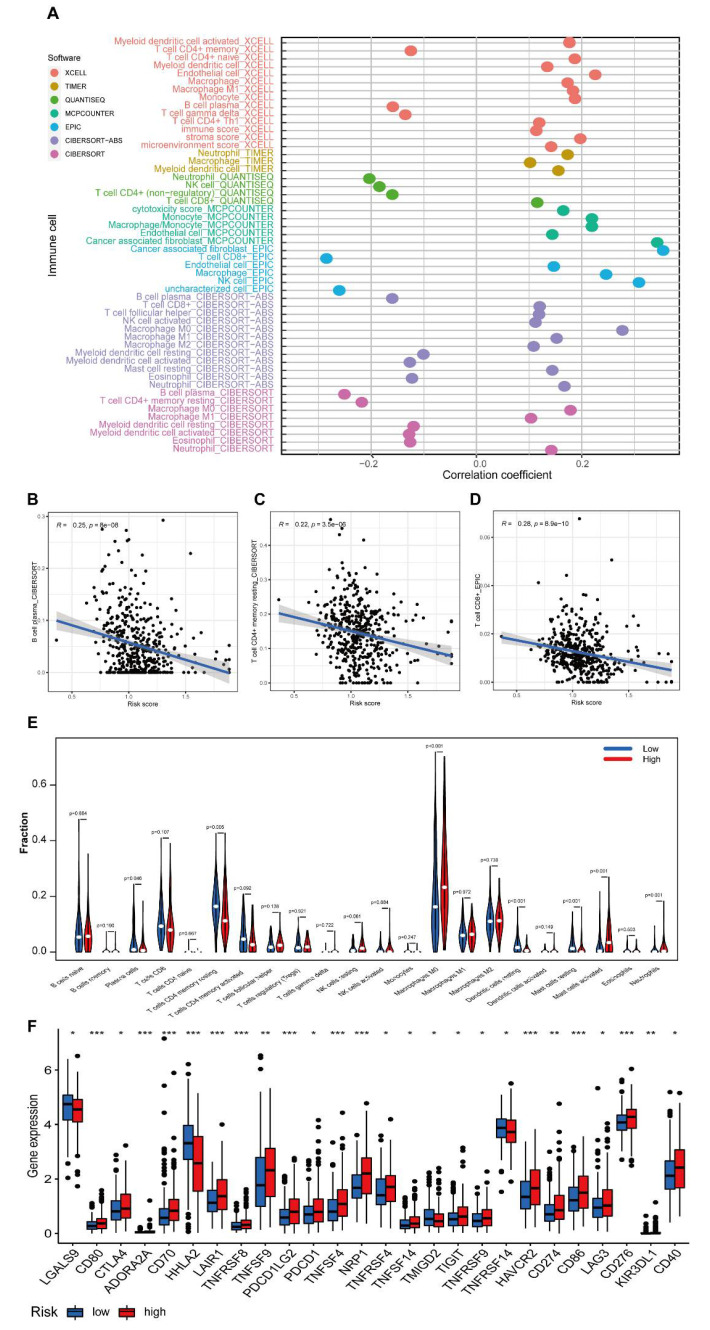
Assessment of the immune landscape in CRC patients. (**A**) A bubble chart shows many immune cells in risk groups using different types of software. (**B**–**D**) Three representative correlations between immune cell levels (B cells, CD4^+^ T cells and CD8^+^ T cells) and the risk scores. (**E)**) Violin plots depicting the difference in the levels of twenty−two immune cells between the two risk subgroups of CRC patients. (**F**) Boxplots illustrating the levels of 26 immune checkpoint genes in two risk subgroups of CRC patients. *: *p* < 0.05, **: *p* < 0.01, and ***: *p* < 0.001, ns: no significance.

**Figure 8 cancers-14-04364-f008:**
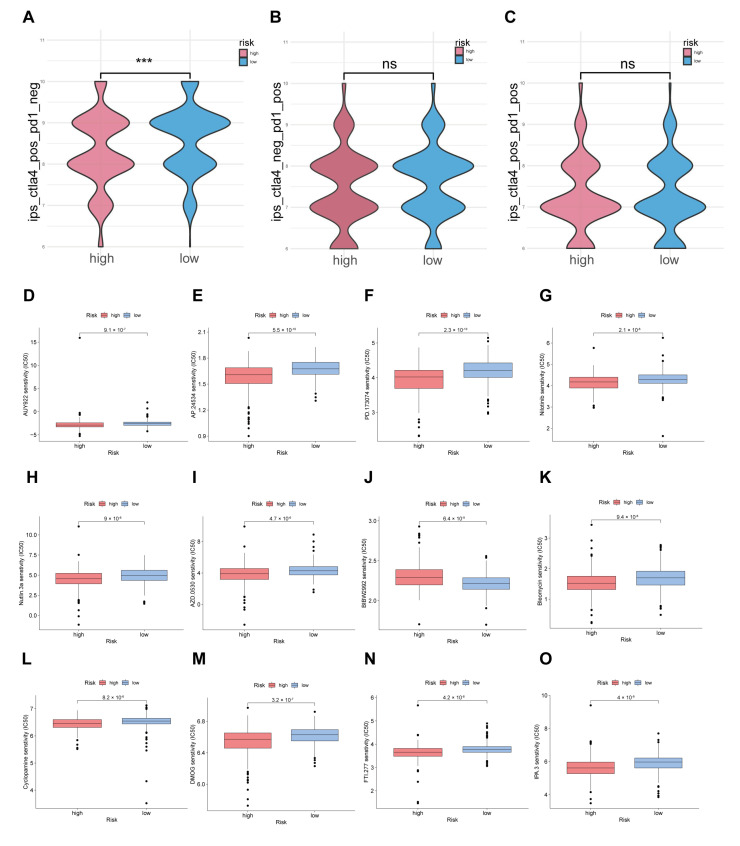
Immunotherapy and drug sensitivity prediction in risk groups of CRC patients. (**A**–**C**) The association between IPS and the risk score of CRC patients based on the TCGA database. (**A**) CTLA4+ and PD1−. (**B**) CTLA4− and PD1+ cells. (**C**) CTLA4+ and PD1+ cells. (**D**–**O**) Boxplot shows the difference in estimated IC_50_ values for 12 chemotherapy drugs, including AUY922, AP.24534, PD.173074, nilotinib, Nutlin.3a Azd.0530, Bibw.2992, bleomycin, cyclopamine, DMOG, FTI.277 and IPA.3, between the two risk groups. ***: *p* < 0.001, ns: no significance.

**Figure 11 cancers-14-04364-f011:**
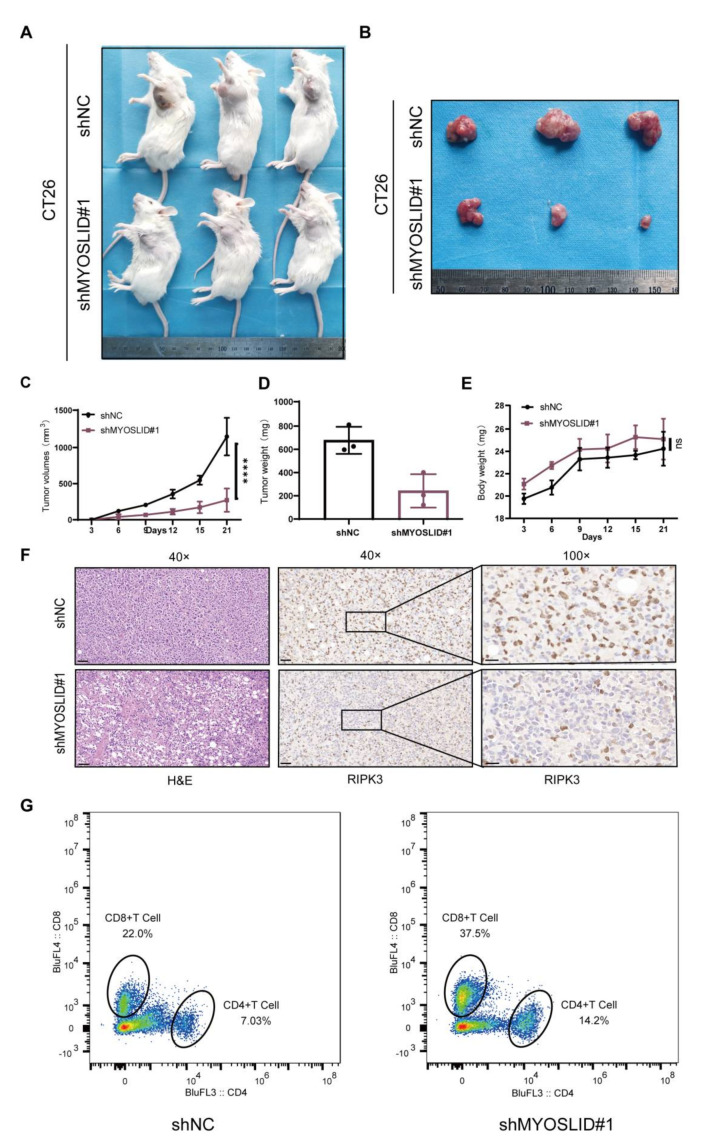
(**A**,**B**) Photos of subcutaneous neoplasia in the shMYOSLID group (**top**) and control group (**bottom**). (**C**,**D**) Tumour volume and tumour weight of subcutaneous neoplasia in the two groups. (**E**) Body weight of mice in each group. (**F**) H&E staining and immunohistochemical staining to detect the expression levels of RIPK3 in subcutaneous neoplasia of each group. (**G**) Flow cytometry analysis revealed significantly lower levels of CD4^+^ T and CD8^+^ T cells in the shMYOSLID group than in the control group.

**Table 1 cancers-14-04364-t001:** The clinical characteristics of CRC patients in the training and testing groups.

Characteristics	Training Group		Testing Group		*p* Value
	No.	%	No.	%	
Age	−		−		−
≤65	89		87		>0.05
>65	123		125		−
Sex	−		−		−
Female	98		98		>0.05
Male	114		114		−
Stage	−		−		−
I	41		32		>0.05
II	84		81		−
III	54		63		−
IV	27		31		−
Pathology T stage	−		−		−
pT1	4		6		>0.05
pT2	42		32		−
pT3	139		151		−
pT4	27		22		−
Pathology N stage	−		−		−
pN0	129		124		>0.05
pN1	49		48		−
pN2	34		40		−
Pathology M stage	−		−		−
pM0	158		157		>0.05
pM1	27		31		—

## Data Availability

The original contributions presented in the study are included in the article/Appendix A. Further inquiries can be directed to the corresponding authors.

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
