# Peer review of "Identification and Validation of the lncRNA MYOSLID as a Regulating Factor of Necroptosis and Immune Cell Infiltration in Colorectal Cancer following Necroptosis-Related LncRNA Model Establishment"

_cancers, 2022, doi:10.3390/cancers14184364_

Round 1
Reviewer 1 Report
In this paper, the authors use bioinformatic approach to identify MYOSLID as a novel LncRNA to regulate necroptosis and tumor microenvironment. The results were further confirmed by clinical, cell and animal experiemnts. The study is novel and the data are clearly presented and supportive. I have two minor points below.
1: The paper need a native speaker or language service to extensively rewrite. There are too many grammar mistakes, I listed some examples below:
Line 68: “regareded” should be “regarded”.
Line 567: “preliminary” should be “preliminarily”.
Lines 569-570; There should be a comma before the word “And”; "my: should be "may".
2: Reference 19, why authors cited an author correction? The original paper should be cited instead.
Author Response
Point 1:1: The paper need a native speaker or language service to extensively rewrite. There are too many grammar mistakes, I listed some examples below:
Line 68: “regareded” should be “regarded”.
Line 567: “preliminary” should be “preliminarily”.
Lines 569-570; There should be a comma before the word “And”; "my: should be "may".
Response1: We are grateful for the reviewer’s valuable comments. The manuscript has been modified professionally by American Journal Experts(AJE). (Verification code 592D-1691-4440-7C75-7A1D). We believe that all grammar errors have been re-corrected, And we hope our manuscript is now qualified for publication. All the change markers have been kept in the revised manuscript.
Point 2: Reference 19, why authors cited an author correction? The original paper should be cited instead.
Response2: Reference 19(Current reference 21) has replaced by citing the original paper.

Reviewer 2 Report
This is an original article concerning necroptosis and Colorectal Cancer
The topic is promising and certainly represents a novelty
Statistical analysis and results are well developed (in particular, the Gene set enrichment analysis). In fact, the authors could have developed two different articles.
Aside from the molecular description, it would be interesting to understand the clinical implications of the present study.
What are the future perspectives?
What is the goal of this type of translational research?
Moreover, I will discuss in the introduction the state-of-the-art of biomakers in CRC (three-four lines).
Targeted therapy for colorectal cancer metastases: A review of current methods of molecularly targeted therapy and the use of tumor biomarkers in the treatment of metastatic colorectal cancer. Cancer. 2019 Dec 1;125(23):4139-4147. doi: 10.1002/cncr.32163. Epub 2019 Aug 21. PMID: 31433498.
Mast Cells, microRNAs and Others: The Role of Translational Research on Colorectal Cancer in the Forthcoming Era of Precision Medicine. J Clin Med. 2020 Sep 3;9(9):2852. doi: 10.3390/jcm9092852
Author Response
Point 1:This is an original article concerning necroptosis and Colorectal Cancer
The topic is promising and certainly represents a novelty
Statistical analysis and results are well developed (in particular, the Gene set enrichment analysis). In fact, the authors could have developed two different articles.
Aside from the molecular description, it would be interesting to understand the clinical implications of the present study.
What are the future perspectives?
What is the goal of this type of translational research?
Response1: We are grateful for the reviewer’s positive comments. The relevant description has been added in lines 57-60.
“The majority of CRC patients cannot benefit from immunotherapy. Therefore, it is imperative to determine how immunotherapy can be augmented in most CRC patients, and it is essential to explore more novel biomarkers in the diagnosis and treatment of CRC.”
In this paragraph, we discussed that it is really important in the future to explore how to make more CRC patients benefit from immunotherapy. And the goal of this type of translational research is that we hope that our newly identified biomarker could be used in clinical someday.
Point 2: Moreover, I will discuss in the introduction the state-of-the-art of biomakers in CRC (three-four lines).
Targeted therapy for colorectal cancer metastases: A review of current methods of molecularly targeted therapy and the use of tumor biomarkers in the treatment of metastatic colorectal cancer. Cancer. 2019 Dec 1;125(23):4139-4147. doi: 10.1002/cncr.32163. Epub 2019 Aug 21. PMID: 31433498.
Mast Cells, microRNAs and Others: The Role of Translational Research on Colorectal Cancer in the Forthcoming Era of Precision Medicine. J Clin Med. 2020 Sep 3;9(9):2852. doi: 10.3390/jcm9092852
Response2: The relevant discussion has been added in lines 52-58
“Although remarkable advances have been made in cancer immunotherapy in recent years and some immunotherapy targets like PD-1, PD-L1 and CTLA4 have been clinically applied[7], it benefits only 3% to 7% of CRC patients who exhibit defective mismatch repair (dMMR) proteins/microsatellite instability (MSI-H) [8][9]. ”
In this paragraph, we discussed that even some novel target biomarkers have been identified. However, only a few CRC patients could benefit from current immunotherapy.
These two papers have been read carefully and cited in this study to illustrate the importance and the meaning of our research. (Citation 6 and Citation8). Thanks for the recommendation.

Reviewer 3 Report
This is an interesting manuscript on the role of long non-coding RNAs in colorectal cancer. The authors have used a broad range of techniques from bioinformatics to animal studies. They have concluded that the lncRNA MYOSLID is a regulating factor for necroptosis and immune cell infiltration in colorectal cancer. The findings advance our understanding of the development of colorectal cancer and present a potential target for intervention. The manuscript is clearly written and would merit publication after consideration of two papers in the literature. The manuscript would benefit from the citation of the paper by Jia et al., entitled “Non-coding RNAs in colorectal cancer: their function and mechanisms” (Frontiers in Oncology 12, article 783079, 2022). This would give the reader background in the field. The authors should also consider the paper by Zhong et al., entitled “A hypoxia-related lncRNA signature correlates with survival and tumor microenvironment in colorectal cancer” (Journal of Immunology Research, article 9935705, 2022). The paper by Zhong et al, reported that MYOSLID could significantly regulate the proliferation, invasion and metastasis of CRC. In contrast to the present manuscript that relates MYOSLID to necroptosis, the paper by Zhong et al. relates MYOSLID to hypoxia related lncRNAs. These are not mutually exclusive relationships and the present manuscript would benefit from a comparison of the findings with those of Zhong et al.
Author Response
Point 1:This is an interesting manuscript on the role of long non-coding RNAs in colorectal cancer. The authors have used a broad range of techniques from bioinformatics to animal studies. They have concluded that the lncRNA MYOSLID is a regulating factor for necroptosis and immune cell infiltration in colorectal cancer. The findings advance our understanding of the development of colorectal cancer and present a potential target for intervention. The manuscript is clearly written and would merit publication after consideration of two papers in the literature. The manuscript would benefit from the citation of the paper by Jia et al., entitled “Non-coding RNAs in colorectal cancer: their function and mechanisms” (Frontiers in Oncology 12, article 783079, 2022). This would give the reader background in the field.
Response1: We are grateful for the reviewer’s valuable suggestions and thanks for all the positive comments about this study. It is truly comprehensive to cite these two papers in our study. The relevant description has been added.
In line 91, we cited the paper entitled “Non-coding RNAs in colorectal cancer: their function and mechanisms” (Frontiers in Oncology 12, article 783079, 2022) as Reference 23, and added the description of the important roles of LncRNA in colorectal cancer and necroptosis.
Point 2: The authors should also consider the paper by Zhong et al., entitled “A hypoxia-related lncRNA signature correlates with survival and tumor microenvironment in colorectal cancer” (Journal of Immunology Research, article 9935705, 2022). The paper by Zhong et al, reported that MYOSLID could significantly regulate the proliferation, invasion and metastasis of CRC. In contrast to the present manuscript that relates MYOSLID to necroptosis, the paper by Zhong et al. relates MYOSLID to hypoxia related lncRNAs. These are not mutually exclusive relationships and the present manuscript would benefit from a comparison of the findings with those of Zhong et al.
Response2: In lines 105-106, we cited the paper entitled “A hypoxia-related lncRNA signature correlates with survival and tumor microenvironment in colorectal cancer” (Journal of Immunology Research, article 9935705, 2022)” as Reference 28. We discussed MYOSLID with its roles in hypoxia condition. And claimed that “the potential role of MYOSLID in regulating necroptosis has never been studied”.

Round 2
Reviewer 2 Report
Congratulations on the work done. I'm satisfied with the changes made